# Development of PCR-Multiplex Assays for Identification of the Herpotrichiellaceae Family and Agents Causing Chromoblastomycosis

**DOI:** 10.3390/jof10080548

**Published:** 2024-08-04

**Authors:** Gabriel S. M. Sousa, Rodrigo S. De Oliveira, Alex B. Souza, Ruan C. Monteiro, Elaine P. T. E. Santo, Luciano C. Franco Filho, Denison L. O. Moraes, Sarah R. De Sá, Silvia H. M. Da Silva

**Affiliations:** 1Programa de Pós-Graduação em Biologia de Agentes Infecciosos e Parasitários, Instituto de Ciências Biológicas, Universidade Federal do Pará, Belém 66075-750, Brazil; denisonlmoraes@outlook.com; 2Laboratório de Micoses Superficiais e Sistêmicas, Seção de Bacteriologia e Micologia, Instituto Evandro Chagas, Ananindeua 67030-000, Brazil; rodrigooliveira@iec.gov.br (R.S.D.O.); alexbritosouza@yahoo.com.br (A.B.S.); elainetavares@iec.gov.br (E.P.T.E.S.); lucianofranco6@gmail.com (L.C.F.F.); sarah.rdsa@aluno.uepa.br (S.R.D.S.); silviasilva@iec.gov.br (S.H.M.D.S.); 3Laboratory of Emerging Fungal Pathogens, Universidade Federal de São Paulo, São Paulo 04023-062, Brazil; rcmonteiro@unifesp.br

**Keywords:** chromoblastomycosis, phaeohyphomycosis, PCR-Multiplex, Herpotrichiellaceae, sequencing, phenotypic, molecular biology

## Abstract

The Herpotrichiellaceae family is an important group of dematiaceous filamentous fungi, associated with a variety of pathogenic fungal species causing chromoblastomycosis (CBM) and phaeohyphomycosis (PHM), both with polymorphic clinical manifestations and worldwide incidence. Currently, the identification of this family and determination of the causative agent is challenging due to the subjectivity of morphological identification methods, necessitating the use of molecular techniques to complement diagnosis. In this context, genetic sequencing of the *Internal Transcribed Spacer* (ITS) has become the norm due to a lack of alternative molecular tools for identifying these agents. Therefore, this study aimed to develop PCR-Multiplex methodologies to address this gap. Sequences from the ITS and *Large Subunit* (LSU) of ribosomal DNA were used, and after manual curation and in vitro analyses, primers were synthesized for the identification of the targets. The primers were optimized and validated in vitro, resulting in two PCR-Multiplex methodologies: one for identifying the Herpotrichiellaceae family and the bantiana clade, and another for determining the species *Fonsecaea pedrosoi* and *Fonsecaea monophora*. Ultimately, the assays developed in this study aim to complement other identification approaches for these agents, reducing the need for sequencing, improving the management of these infections, and enhancing the accuracy of epidemiological information.

## 1. Introduction

The family Herpotrichiellaceae belongs to the phylum Ascomycota, a large group of fungi known for sexual reproduction through the formation of asci and ascospores [1,2]. This family is also part of the class Eurotiomycetes, a fungal group consisting of fungi with diverse morphologies and cosmopolitan distribution, recognized for including species of medical, environmental, and industrial importance, often characterized by their well-defined sexual reproduction structures and resistance to extreme conditions [3].

Within the class Eurotiomycetes, the order Chaetothyriales, which includes the family Herpotrichiellaceae, is characterized by dematiaceous fungi that exhibit dark pigmentation due to the presence of melanin in their cell walls. This melanization provides resistance to various types of environmental stress, such as radiation, high temperatures, and the presence of heavy metals, contributing to the survival and pathogenicity of these fungi [4].

In this context, the family Herpotrichiellaceae comprises five distinct taxonomic clades [5] that include clinically relevant genera such as *Fonsecaea*, *Cladophialophora*, *Exophiala*, and *Rhinocladiella*. These fungi are notable for their ability to cause infections in humans and animals, particularly associated with chronic subcutaneous diseases and systemic infections [6,7,8]. Moreover, various species within this family are thermotolerant, capable of growing over a wide range of temperatures, facilitating their adaptation to different environments and hosts [4].

In this case, the pathogenic species within this family are responsible for causing two different types of infections. The first one is chromoblastomycosis (CBM), a chronic subcutaneous mycosis endemic to tropical and subtropical regions, predominantly affecting rural workers who frequently come into contact with soil and vegetation [8]. It is classified as a neglected tropical disease by the World Health Organization (WHO) due to its prevalence in impoverished areas and the lack of resources for diagnosis and treatment [9], with the main causative agents being species of the genus *Fonsecaea* sp., belonging to the bantiana clade [8,10].

Characterized as an implantation mycosis, chromoblastomycosis (CBM) manifests as verrucous lesions, nodules, and plaques that tend to progress over time to a chronic infection. These lesions generally grow slowly, starting as small papules that can ulcerate and become crusted [11,12]. Patients often experience itching and pain at the site of infection. If left untreated, the disease can lead to secondary bacterial infections, lymphatic dissemination, and significant loss of function in the affected limb due to chronic inflammation and tissue damage [8,10,13].

The other fungal infection frequently associated with species of the family Herpotrichiellaceae is phaeohyphomycosis (PHM), which has global incidence; this pathology is mainly caused by species of the genera *Rhinocladiella* sp., *Exophiala* sp., *Cladophialophora* sp., and *Phialophora* sp. [7]. In humans, PHM may present with a variety of symptoms depending on the site of infection. Cutaneous and subcutaneous infections typically manifest as nodules, abscesses, or ulcerative lesions, often with a dark pigmentation due to the melanin in the fungal cell walls. In mucosal infections, symptoms can include persistent nasal congestion, sinusitis, or oral ulcers [7,14].

When internal tissues are affected, particularly in systemic cases, symptoms may include fever, weight loss, respiratory distress, or neurological deficits, reflecting involvement of organs such as the lungs, brain, or other visceral organs [14,15,16,17]. The disease can be particularly aggressive and fatal in immunocompromised patients if not promptly and effectively treated [7].

Both CBM and PHM cause polymorphic and complex clinical manifestations, often associated with other diseases, making the differential diagnosis challenging [10,14,18]. Additionally, the high phenotypic similarity among species of the family Herpotrichiellaceae makes the identification of the causative agent through morphological analysis highly subjective, primarily dependent on the observer’s experience [19]. From this perspective, there is a need for the application of molecular biology techniques to complement the diagnosis and accurately identify these species.

The identification of these pathogens not only aids in the management of associated infections [11,20] but is also essential for the development of fundamental research to enhance the epidemiological understanding of both PHM and CBM, thus improving the comprehension of the biological diversity within this family and its environmental and parasitic relationships [4,5].

Currently, the primary molecular identification technique for species within the family Herpotrichiellaceae is the genetic sequencing of the *Internal Transcribed Spacer* (ITS) of region of ribosomal DNA (rDNA) [10,21]. This method is typically confined to large research and surveillance centers, impeding access for individuals most affected by these infections to such services. Additionally, it poses challenges for conducting more comprehensive epidemiological studies due to most reports identifying the agent only at the genus level [19].

From this perspective, there is a need for the development of more accessible molecular techniques suitable for laboratories with limited infrastructure, particularly in large endemic regions. Consequently, the present study aimed to develop a molecular methodology based on PCR-Multiplex for the identification of the family Herpotrichiellaceae, the bantiana clade, and the species *Fonsecaea pedrosoi* and *Fonsecaea monophora.*

## 2. Materials and Methods

### 2.1. Clinical Strains

In this study, isolates from the Herpotrichiellaceae family (N = 30) were utilized, comprising the species *Fonsecaea pedrosoi* (N = 22), *Fonsecaea monophora* (N = 5), *Cladophialophora bantiana* (N = 1), *Exophiala dermatitidis* (N = 1), and *Rhinocladiella similis* (N = 1), along with one isolate from the genus *Microascus* sp. (N = 1) of the Microascoceae family. These isolates were maintained in the mycological collection of the Laboratory of Superficial and Systemic Mycoses at the Evandro Chagas Institute (IEC) in the state of Pará, Brazil. The representative fungal agents were originally obtained from clinical samples collected from healthcare units and hospitals within the state of Pará, as documented in Table 1. These agents were previously identified using molecular methods, and their nucleotide sequences were deposited in the GenBank platform of NCBI.

### 2.2. DNA Extraction

The cultures were subcultured in tubes containing YPD Agar and incubated at 30 °C for 14 days for DNA extraction. The extraction process involved collecting approximately 400 mg of fungal mass and adding it to a 2 mL tube containing a solution of 150 μL of lysis buffer (SDS), 150 μL of homogenization buffer, and 150 μL of TE buffer. Glass beads were added, and the microtube was vigorously shaken for 30 min. Subsequently, 15 μL of proteinase K was added, and the microtube was incubated in a water bath at 57 °C for one hour. Afterward, 200 μL of 5 M sodium chloride was added, and it was incubated again at 67 °C for 10 min.

After incubation, 600 μL of the solution was transferred to a new microtube and purified using the phenol–chloroform–isoamyl alcohol protocol described by Campos, 2017 [22]. Subsequently, the Bioflux DNA purification kit (Hangzhou Bioer Technology Co., Ltd., Hangzhou, China) was used according to the manufacturer’s instructions. The extracted DNA was quantified using the NanoDrop 2000© spectrophotometer (Thermo Fisher Scientific Inc.^®^, Waltham, MA, USA). We used the standard value of 1 OD = 50 µg/mL to determine the concentration of double-stranded DNA. Only samples with an OD260/280 ratio between 1.7 and 2.0 were included in the study.

### 2.3. Molecular Identification through Sequencing

Nucleotide sequencing of the ITS region of the isolates was performed to confirm the previously identified species, which include ITS1, 5.8S, and ITS2. For this purpose, amplification of this region was carried out using the primers ITS1(F) TCCGTAGGTGAACCTGCGG and ITS4(R) TCCTCCGCTTATTGATATGC. The PCR conditions consisted of an initial denaturation at 95 °C for 5 min, followed by 35 cycles of denaturation at 94 °C for 1 min, primer annealing at 55.5 °C for 2 min, and extension at 72 °C for 2 min. Finally, an extension phase of 10 min at 72 °C was conducted [23].

The amplification reaction was performed with 4 mM MgCl_2_, 0.4 mM of each dNTP (deoxynucleotide triphosphate), 1 mM of the primers, 0.1 μL of Taq DNA polymerase (Thermo Fisher Scientific Inc.^®^, Waltham, MA, USA), 2.5 μL of 10 mM/L BSA, and 2 μL of DNA, in a final volume of 25 μL. PCR was carried out using a PX2 Thermo Hybaid thermocycler (Artisan Technology Group, Champaign, IL, USA). The amplification product was visualized by agarose gel electrophoresis at 2%.

Following electrophoresis, amplicon purification was performed using the ExoSAP-IT™ PCR Product Cleanup Reagent (Thermo Fisher Scientific Inc.^®^, Waltham, MA, USA), following the manufacturer’s recommendations. Sequencing was carried out using the BigDye™ Terminator v3.1 Cycle Sequencing Kit (Thermo Fisher Scientific Inc.^®^, Waltham, MA, USA), as per the manufacturer’s instructions, and the samples were sequenced on the SeqStudio Genetic Analyzer (Thermo Fisher Scientific Inc.^®^, Waltham, MA, USA). Quality assessment was conducted using the Thermo Fisher Connect Platform “https://www.thermofisher.com/br/en/home/digital-science/thermo-fisher-connect.html (accessed on 10 February 2023)”.

After manual curation, species identification was performed by aligning the sequences with those deposited in the GenBank and ISHAM databases, referencing type strains and considering a similarity value greater than 99% for species determination.

### 2.4. Phylogenetic Analysis

A multifasta file was created using NotePad++ v8.5.4 software, containing the sequences of the strains used in this study and their respective type strains available on the GenBank platform of NCBI. Subsequently, the sequences were aligned using the online Mafft 7 software [24] “https://mafft.cbrc.jp/alignment/server/index.html (accessed on 10 February 2024)’’. For phylogenetic reconstruction, a Web Service of the IQtree software version 2.3.6 [25] “http://iqtree.cibiv.univie.ac.at/ (accessed on 14 February 2024)” was utilized, applying the maximum likelihood method with a substitution model (TNe + G4) and 1000 bootstrap replicates (bt). Visualization and annotation were conducted using the online iTOL software [26] “https://itol.embl.de/ (accessed on 15 February 2024)”.

### 2.5. Primer Design

For primer design, sequences from the Internal Transcribed Spacer (ITS) and Large Subunit (LSU) regions of ribosomal DNA (rDNA) were selected. To achieve this, 100 ITS sequences and 80 LSU sequences representing pathogenic species of the Herpotrichiellaceae family (Appendix A) were retrieved from GenBank. These sequences were separated into two multifasta files and aligned using the online Mafft 7 software [24] “https://mafft.cbrc.jp/alignment/server/index.html (accessed on 10 October 2023)”.

After alignment, manual curation of the sequences was performed using the MEGA 11 software [27]. Nucleotide sequences were selected based on the level of conservation within the following taxonomic groups: (1) Herpotrichiellaceae family; (2) bantiana clade; (3) *Fonsecaea pedrosoi* species; and (4) *Fonsecaea monophora* species. Sequences showing the best conservation according to the mentioned targets were further evaluated for parameters such as melting temperature, % GC content, sequence dimers, etc., and possible sequence incompatibility errors were eliminated using the online oligoanalyzer software from IDT “https://www.idtdna.com/calc/analyzer (accessed on 20 October 2023)”.

Following parameterization, the sequences were validated for their specificity towards the previously mentioned taxonomic targets using the Blastn platform on NCBI “https://blast.ncbi.nlm.nih.gov/Blast.cgi (accessed on 25 October 2023)”, assessing similarity with different taxonomic groups through local alignment. Sequences showing specific alignment to their target were organized into pairs (Forward and Reverse), which were then subjected to in silico PCR evaluation.

### 2.6. In Silico PCR

In silico PCR was applied to validate the specificity of each primer pair in annealing to the target DNA sequences of isolates from the Herpotrichiellaceae family and the bantiana clade, as well as *Fonsecaea pedrosoi* and *Fonsecaea monophora* species, using the Primer-BLAST program “https://www.ncbi.nlm.nih.gov/tools/primer-blast/index.cgi (accessed on 01 November 2023)”. The evaluation parameters used were described by Ye J et al. [28], where the in silico specificity of the primers was assessed to identify potential unintended targets using traceability and specificity rigor as described by Rodrigues et al. [29].

In pursuit of a multiplex PCR approach, only the primer sets (Forward and Reverse) that passed the specificity rigor, forming amplicons of different sizes with melting temperatures close to each other, proceeded to synthesis.

### 2.7. PCR Optmization

Two distinct multiplex PCR assays were conducted: specific primers (1) for the Herpotrichiellaceae family and the bantiana clade; (2) for the species *Fonsecaea pedrosoi* and *Fonsecaea monophora*. In both assays, the FastStart High Fidelity PCR System Kit from ROCHE (Roche, Basel, Switzerland) was used following the manufacturer’s instructions with some modifications. Specifically, 2.5 μL of 10 mM/L BSA was added, and 0.5 μL of each primer at 10 picomoles was used, in addition to 2 μL of DNA at 100 ng/μL, in a final volume of 25 μL.

Both PCRs were conducted using the PX2 Thermo Hybaid thermocycler (Artisan Technology Group, Champaign, IL, USA). For the family–clade multiplex PCR, reaction conditions consisted of an initial denaturation cycle at 95 °C for 5 min, followed by 35 cycles of denaturation at 94 °C for 1 min, annealing at 62 °C for 1 min, and extension at 72 °C for 25 s, with a final extension at 72 °C for 10 min. For the species multiplex PCR, reaction conditions included an initial denaturation at 95 °C for 5 min, 35 cycles of denaturation at 94 °C for 1 min, annealing at 64 °C for 20 s, and extension at 72 °C for 20 s, followed by a final extension at 72 °C for 10 min after the cycles.

The products of both PCR assays were analyzed by agarose gel electrophoresis using 2% UltraPure Agarose (Invitrogen, Waltham, MA, USA) with visualization on the Amersham Imager 600 (GE Healthcare Bio Sciences AB, Uppsala, Sweden).

### 2.8. Assessment of Nonspecific Amplification

Each primer set (Forward and Reverse) was analyzed for selectivity and amplification of unintended targets using samples of human DNA and other causative agents of cutaneous and subcutaneous mycoses such as *Sporothrix* sp. and *Microascus* sp., obtained from clinical samples. The conditions for polymerase chain reaction (PCR) and agarose gel electrophoresis have been previously detailed.

### 2.9. Evaluation of the Minimum Amplification Threshold

For each primer set, the minimum detection limit and amplification of the target DNA were evaluated. Ten-fold serial dilutions were utilized, starting with 100 ng/μL and ending with 0.01 fg/μL. Each primer set had its detection limit assessed individually under single PCR conditions, and visualization occurred through agarose gel electrophoresis as described previously.

## 3. Results

### 3.1. Isolates Used and Phylogenetic Analysis

Based on the phylogenetic analyses of the ITS region sequences of all isolates used in this study and reference strains, a phylogenetic tree (Figure 1) was generated. It was possible to confirm the identification of species within the Herpotrichiellaceae family and distinguish the bantiana clade from other clades.

### 3.2. Primers Designed and In Silico Specificity

At the end of manual curation, the promising sequences for primer synthesis had more than 20 nucleotides in their structure, GC content exceeding 50%, and hairpin formation at temperatures below 30 °C. After in silico specificity evaluation, a primer set (Forward and Reverse) was selected for each desired target (Herpotrichiellaceae family, Bantiana clade, and *Fonsecaea pedrosoi* and *Fonsecaea monophora*), whose annealing regions and primer sizes, along with their sequences, are visualized in Figure 2 and Table 2, respectively.

### 3.3. Testing of the Primers through In Vitro PCR

With the synthesis of the chosen primers after in silico analyses and subsequent optimization of in vitro assays, two distinct PCR-Multiplex assays were created. The first assay successfully amplified only species of the Herpotrichiellaceae family and the bantiana clade in a single reaction, as demonstrated in Figure 3. There was no amplification of nonspecific targets such as *Microascus* sp. and *Sporothrix* sp.

In this context, the formation of two bands of distinct sizes (388 bp and 149 bp) was observed for species belonging to the Herpotrichiellaceae family and the bantiana clade, while species from other clades showed positive amplification only for the family primer, forming a single band (149 bp).

In the second proposed PCR-Multiplex assay, distinct-sized amplicons were generated to identify the species *F. pedrosoi* (479 bp) and *F. monophora* (114 bp). In this context, it was possible to identify these two agents of chromoblastomycosis in a single reaction without cross-reactivity or nonspecific amplification with other species from the same clade, as evidenced in Figure 4.

### 3.4. Detection Limit of the Primers

After evaluating the minimum DNA concentration at which each primer set was capable of efficient amplification, a scale was established as shown in Figure 5, where three out of four primer sets could amplify the target DNA at concentrations of 10 pg/μL. Only the family primers showed lower efficiency, requiring at least 10 ng/μL in a PCR reaction for amplification that could be visualized satisfactorily and was easily discernible.

## 4. Discussion

Infections caused by dematiaceous filamentous fungi are reported on all continents of the world, except Antarctica, with a high prevalence in poor or developing countries, primarily affecting rural workers and immunocompromised individuals [7,8,13,17,30]. In this context, the Herpotrichiellaceae family stands out as an important fungal group, as its species are significant agents of phaeohyphomycosis and chromoblastomycosis [7,10].

Both infections have similar clinical manifestations, tending to differentiate in cases where there is systemic involvement, as CBM is currently characterized only by involvement of cutaneous and subcutaneous tissues, while FEO can disseminate to other tissues and organs of the host [7,31,32].

The initial diagnostic method for both infections is the same, where through direct mycological examination of scrapings or biopsies clarified with Potassium Hydroxide (KOH), it is possible to detect dematiaceous hyphae in the case of FEO and muriform cells for CBM [10,14,33]. Unfortunately, precise identification of the agent presents significant barriers due to the subjectivity of morphological identification methods of species of the Herpotrichiellaceae family, requiring the use of molecular biology techniques such as sequencing of the ITS region to accurately determine the species of the pathogen [11,19,34,35].

The failure to determine these agents directly impacts the management of these infections, as depending on the species, the infection can lead to severe systemic commitment [11,16,36] or be resistant to certain antifungal therapy [37,38,39]. Besides the clinical context, the identification of the agent is crucial for the epidemiological monitoring of these infections, providing data for control and prevention strategies [9].

In this context, the need for complementary molecular biology methodologies that can assist in the correct identification of these species is evident. However, in the current scientific literature, there are no molecular tools available for detecting the Herpotrichiellaceae family or the set of species from the bantiana clade. There are only specific primers designed for certain species, such as *F. pedrosoi* and *Cladophialophora carrioni*, based on PCR-Lamp or Role Circle Amplification [40], as well as some techniques based on Restriction Fragment Length Polymorphism [34].

This scarcity of tools reveals a gap in identification currently covered only by sequencing the ITS region, limiting the identification of these species to large research and surveillance laboratories. Addressing this gap, the present study proposed two new approaches based on PCR-Multiplex, targeting the ITS and LSU regions of rDNA, which are often used as pan-fungal markers [41,42].

These regions of rDNA currently serve as the foundation for molecular identification and phylogenetic analyses of Herpotrichiellaceae species [2,10,43,44], as they are highly conserved and capable of accurately differentiating species within this family, unlike other fungi such as *Sporothrix* sp. [29] and *Cryptococcus* sp. [45].

From this perspective, the first assay accurately identified the entire taxonomic grouping of the Herpotrichiellaceae family and the bantian clade. Determining the family is clinically important as it excludes a series of other agents causing FEO [7,14], limiting the species to specific known agents. In the context of surveillance and research, identifying species within the Herpotrichiellaceae family aids in the development of studies and the monitoring of the diversity and occurrence of these agents in the environment [4,5,46].

On the same scale of importance, identifying the bantian clade delimits the infection to agents of chromoblastomycosis, as this group encompasses all species of the genus *Fonsecaea* sp. [10], responsible for approximately 90% of chromoblastomycosis cases worldwide [8].

In addition to species of the genus *Fonsecaea* sp., it is also possible to identify the species *Cladophialophora bantiana*, an important agent of phaeohyphomycosis [15] within the genus *Cladophialophora* sp., belonging to the same clade as the genus *Fonsecaea* sp. [10,44]. Therefore, there is a need to correlate the results of molecular biology with other morphological analyses or clinical manifestations of the patient, since this species is known for its ability to infect the central nervous system of immunocompromised patients.

The second PCR-Multiplex assay complements the results of the first one, and can be performed immediately after or separately, depending on the objective. In this assay, it was possible to distinguish two of the main agents of chromoblastomycosis (CBM) worldwide, *F. pedrosoi* and *F. monophora*, with the former being responsible for more than 80% of reported CBM cases worldwide [8], while the latter shows a high incidence in Latin America and the Caribbean [21,47,48].

In the clinical context, distinguishing these agents assists in the management of the infection, as there are studies indicating more efficient therapeutic approaches depending on the causative agent of chromoblastomycosis (CBM) [11], which can improve patient prognosis. Patients are typically subjected to lengthy treatments with high chances of recurrence in case of therapeutic failure, increasing the likelihood of treatment abandonment [8].

Furthermore, the identification of these species is crucial for understanding the epidemiology of this infection. According to the latest survey on the global burden of chromoblastomycosis conducted in 2021, most identifications of these agents are only at the genus level [8], revealing a lack of more specific information, compromising more accurate studies on these agents and their host–parasite relationship.

Therefore, both assays were developed to address the gap of lacking molecular identification tools for the evaluated agents, with application designed for analyses from DNA extracted from cultures. However, as no nonspecific amplification was observed with human DNA and other agents of phaeohyphomycosis such as *Microascus* sp. [7,49], which, combined with the low minimum detection limit obtained by the primer sets, except for the one specific to Herpotrichiellaceae, does not exclude the possibility of their application in biological samples, provided that further studies are conducted.

## 5. Conclusions

Our results introduce two new molecular biology methodologies based on the PCR-Multiplex technique. The first aims to identify the Herpotrichiellaceae family and the bantiana clade, while the second targets two major agents of chromoblastomycosis, namely *Fonsecaea pedrosoi* and *Fonsecaea monophora*. In both assays, the amplification of multiple targets aims to reduce the need for multiple analyses and minimize the requirement for sequencing to identify these agents.

In this context, these two new methodologies, besides serving as a method for molecular identification, also enable clinical or research laboratories with limited infrastructure in basic molecular biology techniques to accurately detect these agents, thus contributing to more precise monitoring, which can underpin further research on these species.

## Figures and Tables

**Figure 1 jof-10-00548-f001:**
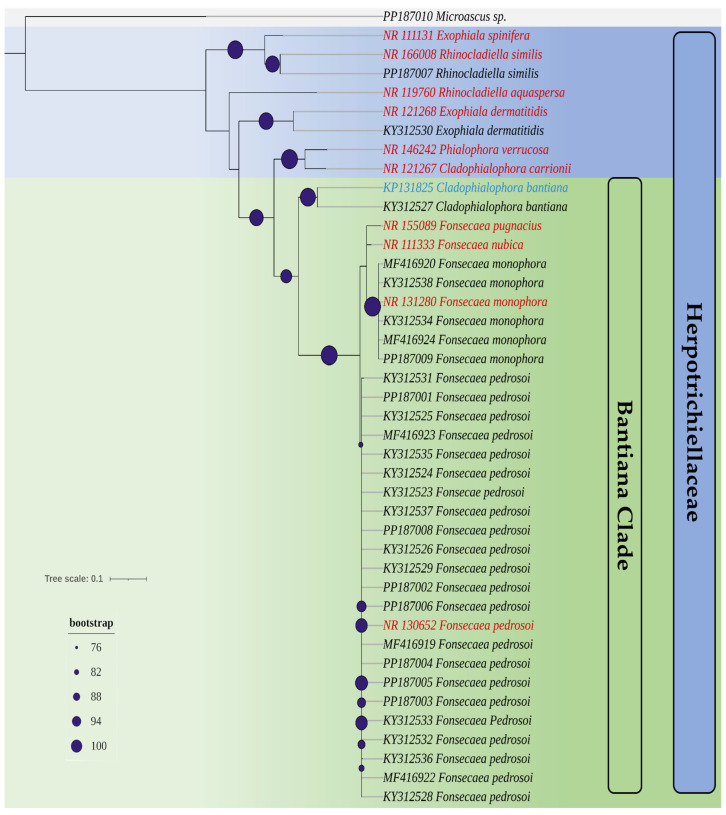
A phylogenetic tree created from the set of ITS sequences of the isolates used in this study and reference strains (highlighted in red and blue). Highlighted with purple circular circles is the bootstrap of the nodes of the genera in the Herpotrichiellaceae family. Highlighted with gray relief is the isolate *Microascus* sp., which does not belong to the Herpotrichiellaceae family and is used as an outgroup.

**Figure 2 jof-10-00548-f002:**
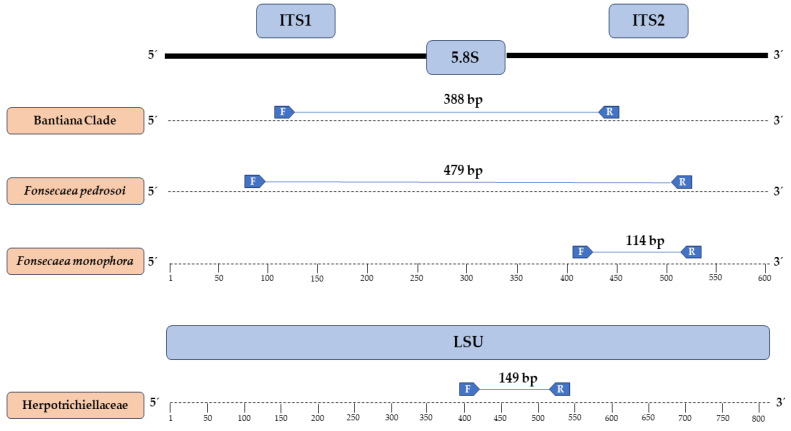
Regions where each chosen primer anneals and their respective amplicons generated in silico.

**Figure 3 jof-10-00548-f003:**
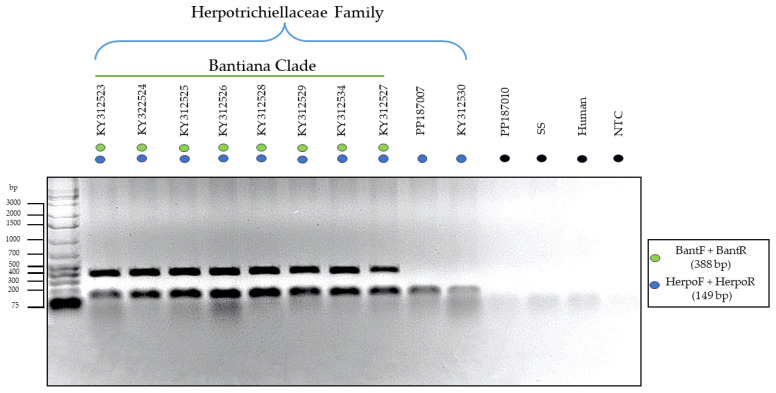
Electrophoresis of in vitro PCR-Multiplex using specific primers for family (blue) and bantiana clade (green). The species where positive amplification occurred for both targets were *Fonsecaea pedrosoi* (KY312523, KY322524, KY312525, KY312526, KY312528, KY312529), *Fonsecaea monophora* (KY312534), and *Cladophialophora bantiana* (KY312527). Meanwhile, species that were positive only for the family were *Rhinocladiella similis* (PP187007) and *Exophiala dermatitidis* (KY312530). There was no amplification of DNA from *Microascus* sp. isolates (PP187010), *Sporothrix* sp. (SS), Human DNA, and No-Template Control (NTC), highlighted in black.

**Figure 4 jof-10-00548-f004:**
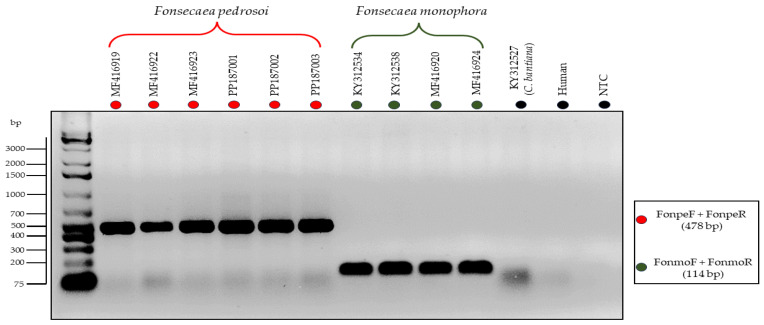
Electrophoresis of in vitro PCR-Multiplex using specific primers for the species *Fonsecaeae pedrosoi* (FonpeF and FonpeR) and *Fonsecaea monophora* (FonmoF and FonmoR). There was no amplification of DNA from *C. bantiana* isolate (KY312527), Human DNA, and No-Template Control (NTC), highlighted in black.

**Figure 5 jof-10-00548-f005:**
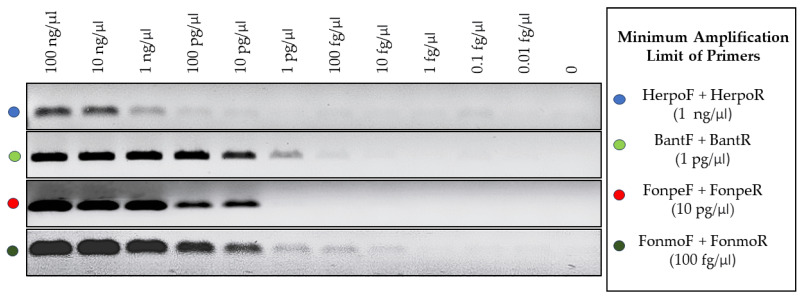
Evaluation of the target DNA detection limit for the 4 primer sets applied in the assays.

**Table 1 jof-10-00548-t001:** Isolates used in this study.

Isolate	Genbank	Species	Anatomical Site	Host	Geographic Origin
IEC-CBM02	KY312523	*F. pedrosoi*	Thigh	Human	Pará/Brazil
IEC-CBM03	KY312524	*F. pedrosoi*	Foot	Human	Pará/Brazil
IEC-CBM04	KY312525	*F. pedrosoi*	Leg	Human	Pará/Brazil
IEC-CBM05	KY312526	*F. pedrosoi*	Leg	Human	Pará/Brazil
IEC-CBM06	KY312527	*C. bantiana*	CNS †	Human	Pará/Brazil
IEC-CBM07	KY312528	*F. pedrosoi*	Hand	Human	Pará/Brazil
IEC-CBM08	KY312529	*F. pedrosoi*	Leg	Human	Pará/Brazil
IEC-CBM09	KY312530	*E. dermatitidis*	*	Human	Pará/Brazil
IEC-CBM10	KY312531	*F. pedrosoi*	Arm	Human	Pará/Brazil
IEC-CBM11	KY312532	*F. pedrosoi*	Leg	Human	Pará/Brazil
IEC-CBM12	KY312533	*F. pedrosoi*	Foot	Human	Pará/Brazil
IEC-CBM13	KY312534	*F. monophora*	Leg	Human	Pará/Brazil
IEC-CBM14	KY312535	*F. pedrosoi*	*	Human	Pará/Brazil
IEC-CBM15	KY312536	*F. pedrosoi*	*	Human	Pará/Brazil
IEC-CBM16	KY312537	*F. pedrosoi*	Thigh	Human	Pará/Brazil
IEC-CBM17	KY312538	*F. monophora*	Leg	Human	Pará/Brazil
IEC-CBM18	MF416919	*F. pedrosoi*	Thigh	Human	Pará/Brazil
IEC-CBM19	MF416920	*F. monophora*	Forearm	Human	Pará/Brazil
IEC-CBM21	MF416922	*F. pedrosoi*	Fist	Human	Pará/Brazil
IEC-CBM22	MF416923	*F. pedrosoi*	Ankle	Human	Pará/Brazil
IEC-CBM23	MF416924	*F. monophora*	*	Human	Pará/Brazil
IEC-CBM5804	PP187001	*F. pedrosoi*	Forearm	Human	Pará/Brazil
IEC-CBM5805	PP187002	*F. pedrosoi*	Hand	Human	Pará/Brazil
IEC-CBM6064	PP187003	*F. pedrosoi*	Leg	Human	Pará/Brazil
IEC-CBM6504	PP187004	*F.pedrosoi*	Arm	Human	Pará/Brazil
IEC-CBM6512	PP187005	*F. pedrosoi*	Foot	Human	Pará/Brazil
IEC-CBM6568	PP187006	*F. pedrosoi*	Arm	Human	Pará/Brazil
IEC-CBM6577	PP187007	*R. similis*	Foot	Human	Pará/Brazil
IEC-CBM6610	PP187008	*F. pedrosoi*	Leg	Human	Pará/Brazil
IEC-CBM6938	PP187009	*F. monophora*	*	Human	Pará/Brazil
IEC-CBM6563	PP187010	*Microascus* sp.	Foot	Human	Pará/Brazil

* Denotes unknown information; † central nervous system.

**Table 2 jof-10-00548-t002:** Sequences of synthesized primers and their respective targets.

Target Species	Primer	Primer Sequence (5′–3′)
Herpotrichiellaceae Family	HerpoF	CTT GCA ACC AGA CTT GAG CGC G
	HerpoR	CGC ATG ACA CCC TGG TCT ATA AGT C
Bantiana clade	BantF	GGC AGG CCC GTC TTA ATC TGA CC
	BantR	GCC GTC ATT GTC TTT AGG AGG GGT G
*Fonsecaea pedrosoi*	FonpeF	CCA ACC CTT TGC TTA CTA GAC CTC
	FonpeR	CCC TTC ATC CGA TAC GTG CTC AA
*Fonsecaea monophora*	FonmoF	GGA CGG CTT GGT GGA GTA AG
	FonmoR	GCC CTT CAT CCG ATA CGT GCT CAG

## Data Availability

Data are contained within the article and Appendix A.

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
