# Peer review of "Development of PCR-Multiplex Assays for Identification of the Herpotrichiellaceae Family and Agents Causing Chromoblastomycosis"

_jof, 2024, doi:10.3390/jof10080548_

Round 1
Reviewer 1 Report
This study developed two PCR-Multiplex methodologies to identify the Herpotrichiellaceae family and the bantiana clade, and determine the species Fonsecaea pedrosoi and Fonsecaea monophora. The assays developed in this study is a complement sequencing. There are still some questions to make it more clear:
1. The author didn't select the genus level Nucleotide sequences, why only made the limited primer pairs identification for family level, clade level and two specific species levels?
2. What is the principle to determine the identify profile? Why didn't include the Cladophialophora species?
3. Did the authors finish the Nonspecific Amplification inter-species of the family? It is not clear shown in the strains list.
1. Is there more comparison for the primer pairs evaluation for the Herpotrichiellaceae family and the bantiana clade?
Author Response
Caro revisor,
Por favor, verifique o anexo.

Reviewer 2 Report
first, there are PCR + sequencing with marker ITS 1-4 or ssu to identify those family to species level, what is the aim to do this research?
second: bantiana clade: what do you mean?
why detect those at family level? clade level?
introduction and discussion sections deviated to other direction but methodology.
fig.1: bantiana clade you marked is wrong.
Author Response
Dear Reviewer 2,
Please see the attachment.

Reviewer 3 Report
I do not have major comments for this paper.
1- I would have been improtant to add a greater number of strains for microorganisms with very few strains. I am not sure if the authors could request more strains from the other researchers.
2. It is true that the phenothypic identification of these microorganisms is difficult and subjetive and that molecular biology procedures can help in precise identification. However, the feasibility of having enough equipment in developing countries is low. Please, justify the imminent use of this equipment for an identification that is related to a specific treatment according the etiological agent.
Author Response

(The authors gave the same response as above.)
